# Influence of the Loading Rate on the Cracking Process of Concrete in Quasi-Static Loading Domain

**Pierre Rossi**

MAST-EMGCU, Université Gustave Eiffel, IFSTTAR, F-77447 Marne-la-Vallée, France; pierre.rossi@univ-eiffel.fr

**Abstract:** This study presents analysis of two types of experimental test related to the crack propagation in concrete specimens subjected to high-sustained loading levels and quasi-static loadings. The concept of the equivalent crack length is introduced to perform this analysis. Even though this analysis is partial, it shows the influence of loading rate conditions on the crack process rate. This result shows that, in the domains of low and very low loading rates, the concrete mechanical characteristics linked to the cracking process (for example, tensile strength, post-cracking behaviour, etc.) are dependent on the loading rates applied to the specimens for determining them.

**Keywords:** concrete; sustained loadings; quasi-static loadings; crack propagation rate





## 1. Introduction

Previous research related to the influence of the loading rate (in the framework of quasi-static loadings) on the mode I critical intensity factor, $K_{IC}$, demonstrated that this cracking parameter decreased linearly with the crack propagation rate [1,2]. In this experimental study (called, in the present work, the first experimental study) performed on a very large double cantilever beam (DCB), adhering to the conditions of the applicability of linear elastic fracture mechanics (LEFM) theory to concrete [1–13], the equivalent crack concept was used (see Chapter II). The domain of loading rates considered was low. Indeed, the fracture mechanics test was performed under imposed notch opening rates that varied between 4 and $33.5 \times 10^{-4}$ mm/s. The concrete studied had a compressive strength of 54.5 MPa and contained a larger aggregate size of 12 mm.

In a more recent experimental study (called the second experimental study) on crack propagation of pre-cracked beams subjected to sustained loadings [14], it was demonstrated that the evolution of the deflection under high sustained loading levels ($\geq$75%) was the consequence of crack propagation. In this experimental study, the evolution of the specimen compliance was considered with the evolution of the notch opening rate. This notch opening rate was $\leq 2.5 \times 10^{-5}$ mm/s, so a lot less than in the first experimental study. In consequence, it is relevant to consider that the crack propagations under high-sustained loading are similar to the crack propagations under very low imposed notch opening rates.

The objective of the present work is to determine the evolution of the crack propagation rate related to the second experimental study and to compare it to the crack propagation rate evolution observed in the first experimental study.

By performing this new analysis of the two previous experimental studies, the more general objective of the present work is to propose a global vision of strain rate effects under very low and low loading rates.

## 2. Equivalent Crack Concept

In materials, such as concrete, it is well known that macrocrack propagates within the microcracking zone at its front tip, commonly called the fracture process zone [1,15–19].

It is possible to follow the propagation of a single macrocrack in a concrete structure through the elastic compliance evolution of this structure [1,10,12,13,20–24].

In practice, several steps have to be performed to determine this crack propagation from the knowledge of the compliance evolution:

First step: the theoretical evolution of the elastic compliance.

The theoretical evolution of the elastic compliance of a given specimen in function of the crack length has to be determined numerically. In order to do that, idealized cracks of different lengths are considered in the framework of finite element simulations. Hence, different finite element meshes are considered for these numerical simulations. Each mesh is related to one idealized crack length. In each new mesh, the idealized crack length is simulated by using classical interface elements considered as not "active". It means that these interface elements have no rigidity and no resistance to permit the crack opening without consuming any energy. Then, elastic calculations are performed for each idealized crack length. It is important to note that to get a good evaluation of the elastic kinematic field around a crack a very fine mesh has to be used (to model the stress concentration around the crack).

Furthermore, straight crack propagation is assumed.

A function (a polynomial trend line) for best fitting the theoretical elastic compliance results versus the idealized crack lengths can be determined after calculations have been performed for five or six different idealized crack lengths.

Second step: equivalent crack length ($a_e$) propagation.

The experimental evolution of an ***equivalent crack length*** may be determined knowing both the experimental evolution of the elastic compliance of the specimen and the theoretical relation between the elastic compliance and the idealized crack length. The term *equivalent* is used to express the fact that it exists at an idealized crack (numerical crack) length, which is mechanically equivalent to the real macrocrack with its imperfect/tortuous path and its fracture process zone (at its front tip).

Hence, the equivalent crack is an idealized crack that has the same global mechanical effect (compliance of the specimen) as the real tortuous crack with its process zone.

## 3. Rate Propagation of the Equivalent Crack Length in the Second Experimental Study

### 3.1. Information on the Experimental Study

The following section intends to provide a brief summary of the experimental study, fully detailed in reference [14].

Beams of 200 mm × 150 mm × 700 mm were used to perform 4-point bending tests with a 200 mm span between the loading points and a 600 mm span between the beam supports.

The specimens were unmolded 24 h after casting. They were protected from drying by two layers of auto-adhesive aluminum tape and stored in a controlled temperature and humidity room.

The concrete studied had a compressive strength of 40 MPa and contained larger aggregates size of 20 mm.

The beams under high-sustained loading level were tested using a 100 kN hydraulic machine. The mid-span deflection and the horizontal displacement (D25) at 25 mm on the tensile bottom face of the beam were continuously monitored throughout the tests.

In order to pre-crack the specimens, the adopted strategy was the following: a deflection value, $\delta_0$, related to the post-peak behaviour was chosen from the average bending stress versus the deflection curve of the static tests. This value, $\delta_0$, corresponds to a certain macrocrack length and the loading level, called $P_0$. It was chosen to vary $\delta_0$ between 0.08 and 0.15 mm to study its influence on the possible coupling between crack propagation and the delayed behaviour of concrete under sustained loading. After the creation of the initial crack, the specimens were unloaded. From this point onwards, the specimens were reloaded to a force, corresponding to a certain percentage of $P_0$ on the same testing apparatus. This force was called $P_S$, as illustrated in Figure 1.

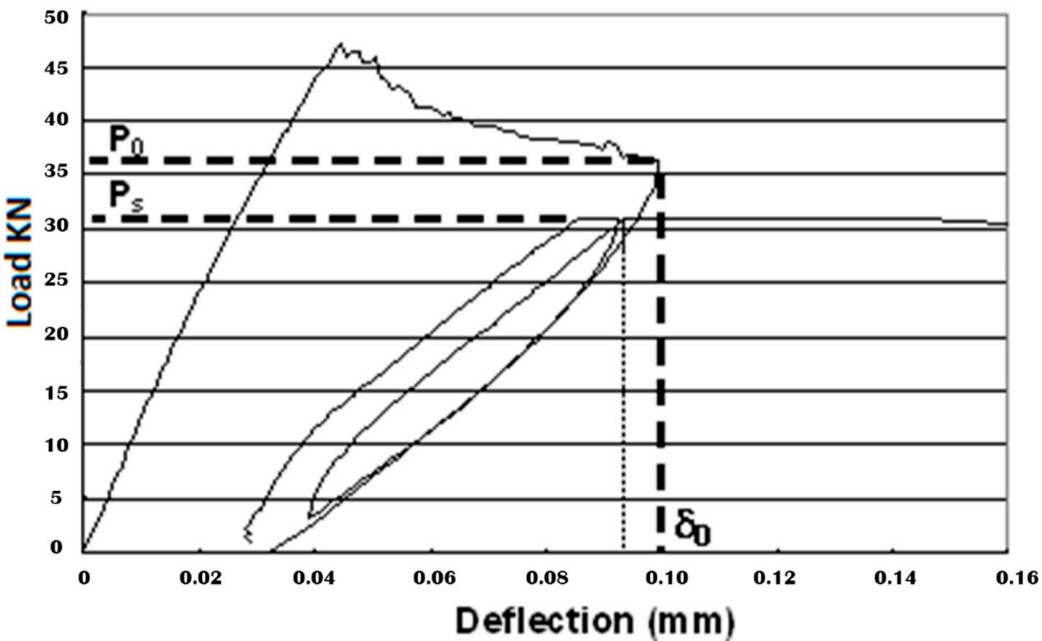

**Figure 1.** Load-deflection curve—procedure adopted for the sustained loading.

Unloading/reloading cycles were performed to evaluate the elastic compliance evolution of the specimen. The elastic compliance was calculated from the load versus mid-span deflection curve using the linear segment of the reloading curve of each cycle, as shown in Figure 2. The maximum duration of the sustained loading on the testing device did not exceed three days in order to avoid thermal effects related to the use of a hydraulic system, after which the tests were stopped.

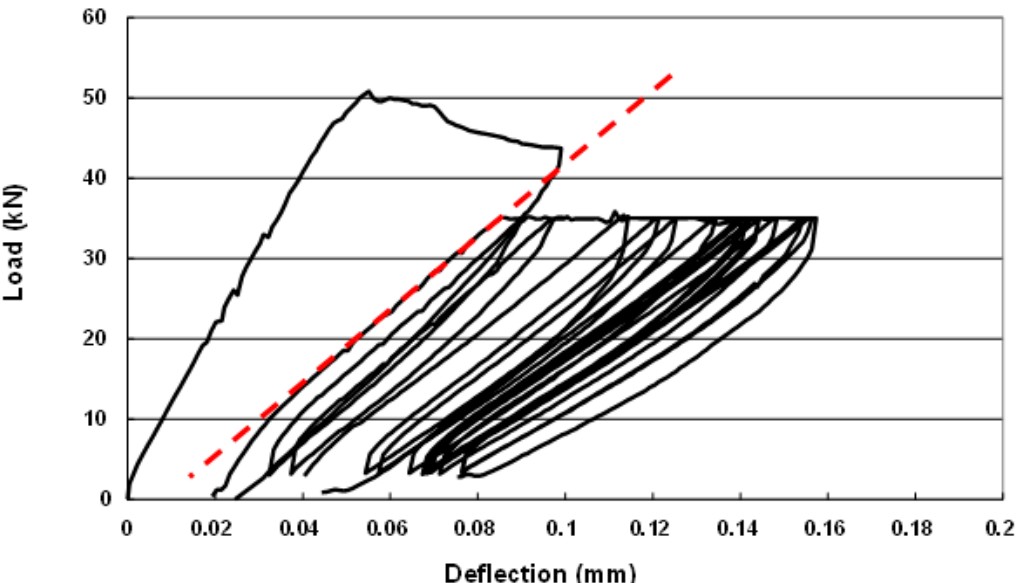

**Figure 2.** Example of unloading/reloading cycle procedure during sustained loading.

Table 1 summarizes the sustained loading level, $P_S/P_0$; the deflection at mid-span, corresponding to the pre-crack level, $\delta_0$; the failure time; the number of unloading/reloading cycles; and the secondary creep displacement (D25) rates [14].

**Table 1.** Information about the secondary displacement (D25) creep rates related to the specimens tested.

| Specimen n° | $P_S/P_0$ | $\delta_0$ (mm) | Failure Time (s) | Cycles Number | Secondary Displacement (D25) Creep Rate ($\times 10^{-7}$ mm/s) |
|---|---|---|---|---|---|
| 1 | 0.85 | 0.1 | 4400 | 4 | 86.5 |
| 2 | 0.85 | 0.08 | 1874 | 3 | 231 |
| 3 | 0.80 | 0.08 | 72,716 | 6 | 2.7 |
| 4 | 0.76 | 0.1 | No failure—stopped the test 252,594 | 8 | 0.311 |
| 5 | 0.80 | 0.1 | 93,713 | 12 | 6.15 |

*Recall*: When creep behaviour of a given concrete specimen is concerned, this behaviour can be analyzed considering the evolution with time of different displacements on the specimen (specimen deflection, for example). Generally, all these evolutions follow three steps: a first step with a decreasing displacement rate, called primary creep; a second step with a constant displacement rate, called secondary creep; and a third step with a strong increase in the displacement rate until the rupture of the specimen. This last step is called tertiary creep.

### 3.2. Equivalent Crack Propagation

An example of a finite element mesh used to determine the theoretical elastic compliance versus idealized crack length curve (see Section 2) is presented in Figure 3.

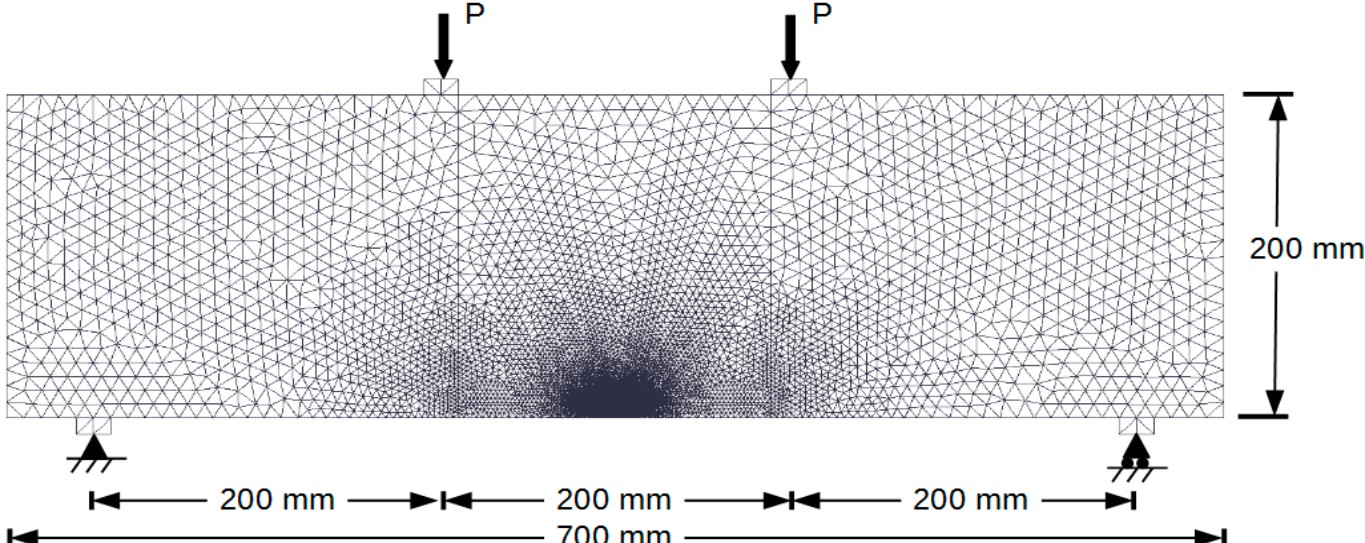

**Figure 3.** Example of finite element mesh used for determining the theoretical relation between the compliance of the beam and the crack length.

The 2D simulations in plane stress conditions are performed. Triangular non-linear elements are used.

The theoretical elastic compliance versus the idealized crack length curve is presented in Figure 4.

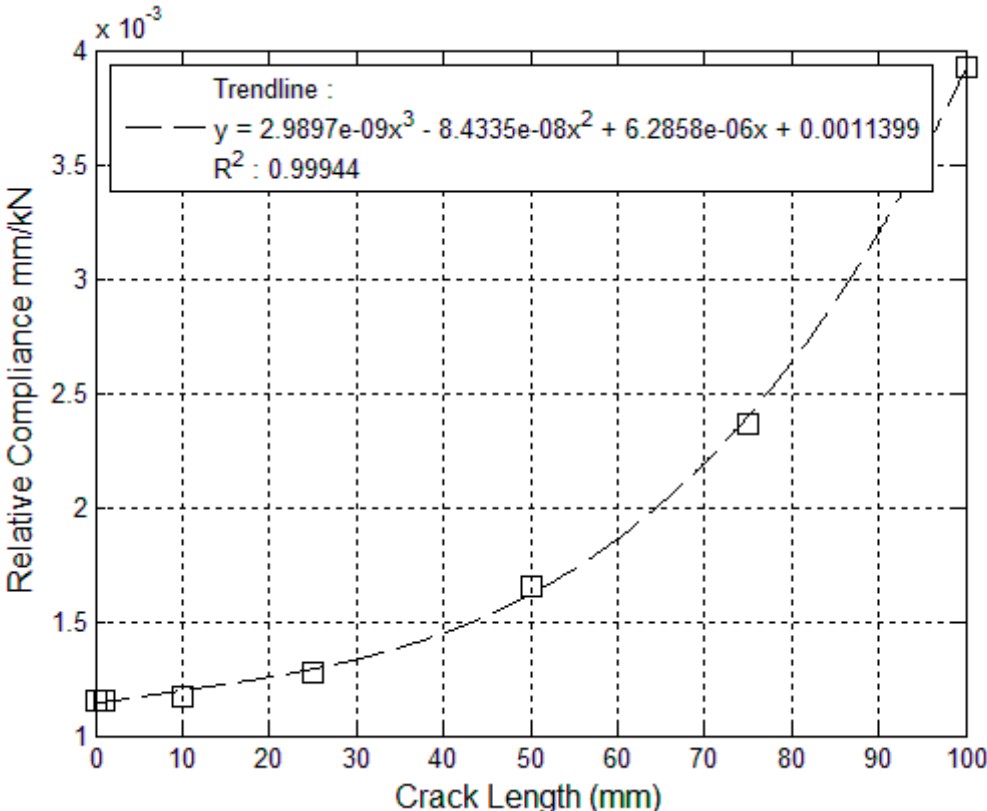

**Figure 4.** Beam compliance versus crack length theoretical curve.

The trend line best fitting the results follows the polynomial relation:

$$C(a) = 2.99 \times 10^{-9} \times a^3 + 8.44 \times 10^{-8} \times a^2 + 6.29 \times 10^{-6} \times a^2 + 1.14 \times 10^{-3} \quad (1)$$

where C (in mm/kN) is the theoretical elastic compliance and a (in mm) is the idealized crack length.

Given the above polynomial relation and the experimental evolution of the elastic compliance related to each test, equivalent crack lengths were calculated at time intervals corresponding to the unloading/reloading cycles for each test and summarized in Table 2.

**Table 2.** Equivalent crack length versus experimental compliance for each test.

| Specimen n° | Time (s) | Experimental Compliance ($\times 10^{-3}$ mm/kN) | Equivalent Crack Length (mm) |
|---|---|---|---|
| 1 | 1511 | 2.7 | 81 |
|  | 2128 | 2.8 | 83 |
|  | 2802 | 2.9 | 84 |
|  | 3976 | 3.1 | 88 |
| 2 | 705 | 2.6 | 80 |
|  | 1281 | 2.8 | 83 |
|  | 1753 | 3.0 | 87 |
| 3 | 2299 | 2.5 | 78 |
|  | 6448 | 2.7 | 82 |
|  | 15,240 | 3.0 | 86 |
|  | 66,338 | 2.9 | 85 |
|  | 68,508 | 3.0 | 87 |
|  | 72,358 | 3.3 | 91 |

**Table 2.** *Cont.*

| Specimen n° | Time (s) | Experimental Compliance ($\times 10^{-3}$ mm/kN) | Equivalent Crack Length (mm) |
|---|---|---|---|
| | 3349 | 3.2 | 90 |
| | 13,332 | 3.4 | 94 |
| | 17,195 | 3.3 | 92 |
| 4 | 20,802 | 3.6 | 96 |
| | 24,231 | 3.7 | 97 |
| | 250,407 | 3.8 | 99 |
| | 252,022 | 4.1 | 102 |
| | 252,594 | 4.1 | 102 |
| | 4388 | 3.2 | 90 |
| | 7548 | 3.3 | 91 |
| | 10,258 | 3.4 | 92 |
| | 14,691 | 3.4 | 92 |
| | 18,053 | 3.5 | 95 |
| | 21,992 | 3.6 | 96 |
| 5 | 25,384 | 3.6 | 96 |
| | 28,774 | 3.7 | 97 |
| | 32,089 | 3.9 | 99 |
| | 85,274 | 3.9 | 100 |
| | 88,228 | 3.9 | 100 |
| | 91,922 | 4.1 | 103 |

Figures 5 and 6 present the equivalent crack propagation versus time curves. As this study aims at providing information regarding the crack propagation rate under sustained loading, the crack propagation is considered starting at t = 0 s, which corresponds to the first unloading/reloading cycle.

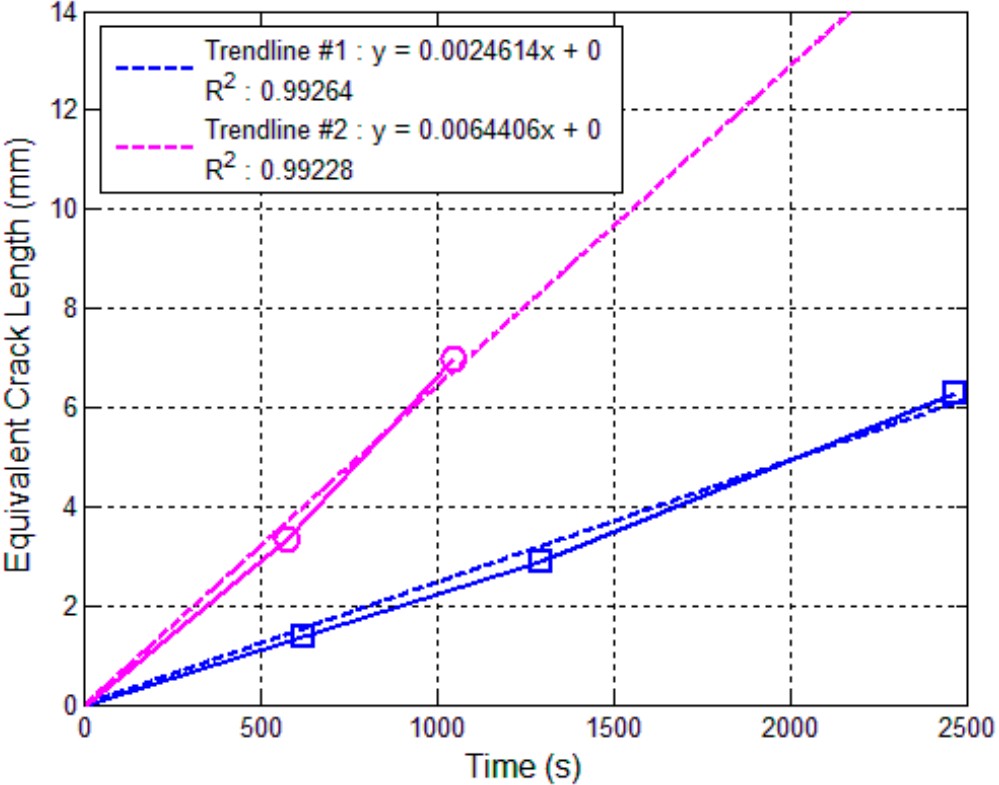

**Figure 5.** Equivalent crack length versus time curves related to the creep loading level of 85%.

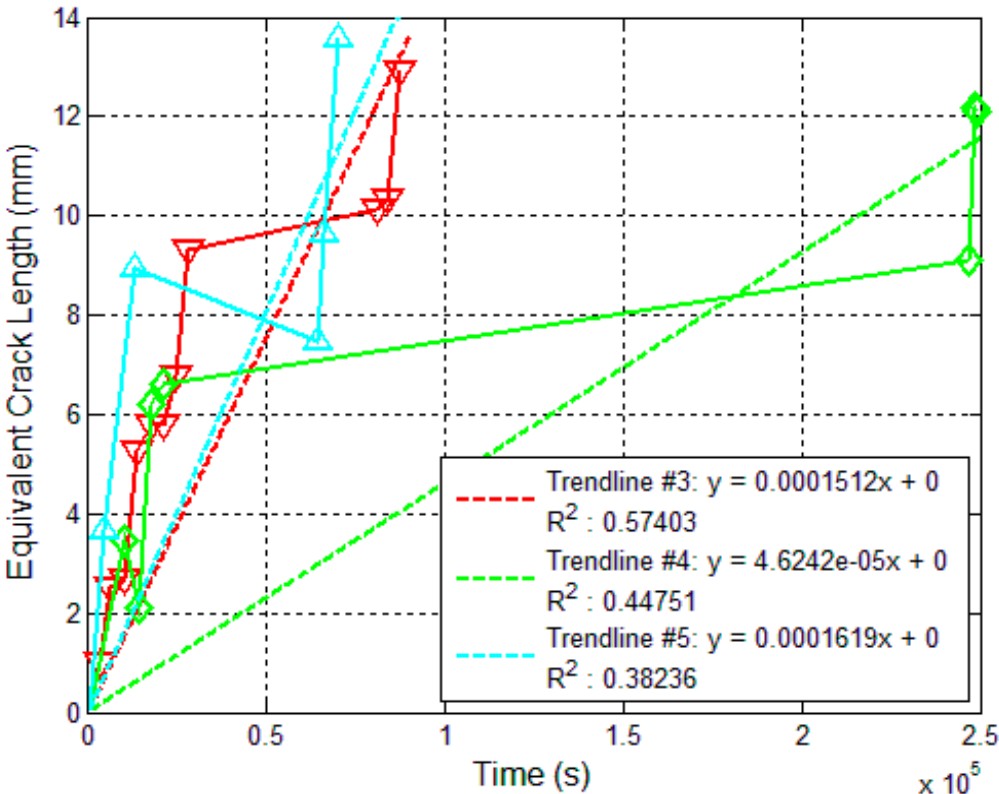

**Figure 6.** Equivalent crack length versus time curves related to creep loading levels of 75 and 80%.

If Figure 5 shows that it is easy to determine the propagation rate of the equivalent crack when the loading level is 85%, Figure 6 shows that it is much more difficult when this loading level is between 75 and 80%. This observation can be explained easily by the fact that the strong heterogeneity of concrete plays a strong role when the crack propagation rate is very low. Indeed, in this situation, the crack can be delayed for a certain time when it encounters stronger zones.

Despite the strong perturbation of the curves in Figure 6, it was still decided to determine the equivalent crack propagation rates for this domain of loading levels. The trend lines, presented in this figure, have to be obviously considered as a kind of "average behaviour".

### 3.3. Propagation Rate of the Equivalent Crack Length

Figure 7 presents the propagation rate of the equivalent crack versus the secondary creep displacement (D25) rate. It shows that a linear relationship exists between the propagation rates of the equivalent crack and the secondary creep displacement rates.

Figure 7 also demonstrates that the propagation rates of the equivalent crack related to loading levels less than 80% are so low that a very strong approximation is made in their determinations. It can be noted that the secondary displacement (D25) creep rate can be assimilated to a notch-opening rate because the difference between these two parameters is not very important.

Hence, by considering Figure 7, it is easy to determine the following relationship between the crack propagation rate and the notch-opening rate:

$$\dot{a}_e = 280 \, \dot{v} \; (\mathrm{mm/s}) \tag{2}$$

when $2.5 \times 10^{-5}$ mm/s and $0.75 \leq PS/P0 \leq 0.85$, $\dot{v}$ being the notch-opening rate.

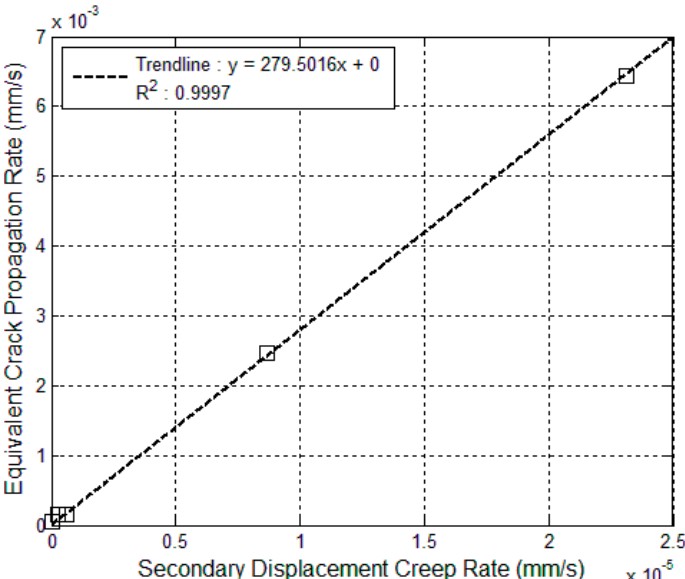

**Figure 7.** Equivalent crack propagation rate versus secondary displacement (D25) creep rate curve.

Equation (2) is surely valid for $\hat{v}$ values larger than $2.5 \times 10^{-5}$ mm/s (when $0.85 < P_S/P_0 \leq 1$).

## 4. Discussion

In the framework of the first experimental study [1,2], only the linear relationship between $K_{IC}$ and the equivalent crack rate was determined. However, it is easy from this paper to determine the relation between the equivalent crack rate and the notch-opening rate imposed during the study. Hence, it is a new information proposed in the present work.

By doing that, the following relation is found:

$$\dot{a}_e = 590 \, \dot{v} \, (\text{mm/s}) \tag{3}$$

when $4 \leq \hat{v} \leq 33.5$ ($\times 10^{-4}$ mm/s).

It is important to point out that Equations (2) and (3) are only related to one type of concrete (one type of mix design) and one type of mechanical test each. Hence, they cannot be considered as general relations.

The most interesting thing to note is that they permit the making of quantitative comparisons between the crack propagation rate related to very low and low loading rates. Indeed, the strong difference observed between these two rates cannot be explained only by the difference between the concrete and the test.

Considering these remarks, it can be pointed out that, in all likelihood, a notch opening rate domain exists between 2.5 and $40 \times 10^{-5}$ mm/s that ensures the transition between the two domains of notch opening rate dependency defined in Relations (2) and (3).

So, it is interesting to try to understand why these three domains of notch opening rate dependency could exist.

For that, it is necessary to consider different physical mechanisms that could intervene during crack propagation in concrete in the quasi-static domain of loading. It means in crack-opening rate domains where inertia effects and/or viscosity effect of water does not exist, these physical phenomena exist for higher crack-opening rates [25–28].

There are two main physical mechanisms, well detailed in previous papers [14,25,29], that can be considered in these crack-opening rate domains.

When a macrocrack propagates within its process zone, there are transfers of water and water vapor, inside the process zone, from the cement paste porosity to the microcracks. These fluid transfers lead to a decrease in the size of the water menisci existing in the

cement paste porosity. This decrease leads to a strong increasing of the Laplace forces (superficial tensions) in the process zone.

The water coming into the microcracks of the process zone has a very well-known chemical consequence: the self-healing of these microcracks.

The superficial forces and the self-healing induce an autogenous shrinkage inside the process zone that introduces compressive stresses. These compressive stresses resist the crack propagation.

The more significant the crack-opening rate is, the less the superficial forces and the self-healing of the microcracks can occur and the smaller the compressive stresses opposing the crack propagation are. It is the reason why the crack propagation rate increases with the crack-opening rate.

The reason why three domains of crack-opening rate dependency can exist in the case of quasi-static mechanical loading is the following: the fluid transfers (physical phenomenon) and the self-healing (chemical phenomena) have not the same kinetic. Fluid transfers are faster than self-healing of microcracks. Therefore, it is possible that, for a certain crack opening rate, self-healing phenomena disappear, and fluid transfers continue to exist.

Hence, to summarize the assumptions proposed concerning the three domains of crack-opening rates dependency and the physical and chemical phenomena associated with them, it can be said that:

*First domain (Equation (2))*: the existence of superficial forces and the self-healing of microcracks decreasing with the increase in the crack-opening rate;

*Second domain (intermediate domain)*: weak existence of self-healing with the predominance of superficial forces;

*Third domain (Equation (3))*: only existence of superficial forces that decreases with the increase in the crack-opening rate.

To conclude this study, it could be interesting, in the framework of future experimental work, to perform fracture mechanics tests for studying the notch-opening rate domain between 1 and $500 \times 10^{-5}$ mm/s. Several concrete mix designs could be considered. This study would determine whether the assumption of the existence of the three domains of the crack-opening rate dependency proposed in this study should be confirmed.

## 5. Conclusions

This study presents analysis of two types of experimental tests related to the crack propagation in concrete specimens subjected to high-sustained loading levels and quasi-static loadings.

The concept of equivalent crack length is introduced to perform this analysis.

This equivalent crack is an idealized crack (numerically modeled by no active interface elements), which has the same global mechanical effect (compliance of the specimen) as the real tortuous crack with its process zone.

Even though this analysis is partial, it shows the influence of loading rate conditions on the crack process rate.

This result tends to show that, in the domains of low and very low loading rates, the concrete mechanical characteristics linked to the cracking process (as an example, tensile strength, post-cracking behaviour, etc.) are dependent on the loading rates applied to the specimens for determining them.

It will be necessary, in the future, to confirm this analysis by performing more fracture mechanical tests on different concretes.

**Funding:** This research received no external funding.

**Data Availability Statement:** Not applicable.

**Conflicts of Interest:** The author declares no conflict of interest.

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
