# Peer review of "Influence of the Loading Rate on the Cracking Process of Concrete in Quasi-Static Loading Domain"

_2673-4109, doi:10.3390/civileng4010001_

Round 1

Reviewer 1 Report

The article is very interesting and presents some important information. However, there are some minor issues that should be corrected before publication.

1. the text should be revised. there are some inconsistencies. For example, in page 8, lines 160-162 it looked like an incomplete sentence.

2. The trend lines in Figure 5, with just 3 points are meaningless (we can see that there is a linear correlation but presenting the r2 with just 3 points...)

3. There is obviously no linear correlation in Figure 6. The trend lines are just a kind of "average" behavior

4. There is no information regarding concrete composition and properties.

Author Response

Dear Sir,

All your remarks and proposal have been considered in the revised paper

Reviewer 2 Report

This paper presented a combined experimental and numerical study on the influence of the loading rate on the cracking process of concrete under quasi-static loading based on the equivalent crack length concept. The obtained results are genuine, initiative, extensive and practically useful, and the discussion and conclusions are comprehensive, sound and convincing. The paper was fairly well prepared, including tables, figures and references. The paper provides much new and useful information so it is worthwhile to publish. There are some technical and editorial issues which need to be further revised before the paper can be published. The reviewer has clearly marked the comments and suggestions in the PDF manuscript submitted by the authors. The authors should pay attention to each of these by marking the revisions clearly in the revised manuscript in different colours for quick re-review.

Author Response

Dear Sir,

All your comments and proposals have been considered in the revised paper.

Reviewer 3 Report

The paper is good and valuable in its different parts. It can be accepted for publication

Author Response

Thank you for accepting my paper
